# Opportunities for Regulatory Authorities to Assess Animal-Based Measures at the Slaughterhouse Using Sensor Technology and Artificial Intelligence: A Review

**DOI:** 10.3390/ani13193028

**Published:** 2023-09-26

**Authors:** Annika M. Voogt, Remco S. Schrijver, Mine Temürhan, Johan H. Bongers, Dick T. H. M. Sijm

**Affiliations:** Office for Risk Assessment & Research (BuRO), Netherlands Food and Consumer Product Safety Authority (NVWA), P.O. Box 43006, 3540 AA Utrecht, The Netherlands

**Keywords:** animal welfare, meat inspection, camera surveillance, sensors, abattoir, machine learning, precision livestock farming, innovation

## Abstract

**Simple Summary:**

Various measurements can be obtained on an animal. These measurements can provide valuable information on animal welfare. Sensors and smart algorithms can automatically perform those measurements and aid in achieving more automated meat inspection and welfare assessment at the slaughterhouse. This study provides an overview of animal welfare measurements at the slaughterhouse and gives examples of the available technologies to record and use these measurements. There are several technologies available, either applied in research, on farm or at the slaughterhouse. However, according to current European Union (EU) law the meat inspection must be performed by a veterinarian. Therefore, sensor technology cannot yet replace a human during the meat inspection, but it has potential to play an important role in the future. Currently it can already add value to the inspections and provide better insight into animal welfare issues than by human inspections alone.

**Abstract:**

Animal-based measures (ABMs) are the preferred way to assess animal welfare. However, manual scoring of ABMs is very time-consuming during the meat inspection. Automatic scoring by using sensor technology and artificial intelligence (AI) may bring a solution. Based on review papers an overview was made of ABMs recorded at the slaughterhouse for poultry, pigs and cattle and applications of sensor technology to measure the identified ABMs. Also, relevant legislation and work instructions of the Dutch Regulatory Authority (RA) were scanned on applied ABMs. Applications of sensor technology in a research setting, on farm or at the slaughterhouse were reported for 10 of the 37 ABMs identified for poultry, 4 of 32 for cattle and 13 of 41 for pigs. Several applications are related to aspects of meat inspection. However, by European law meat inspection must be performed by an official veterinarian, although there are exceptions for the post mortem inspection of poultry. The examples in this study show that there are opportunities for using sensor technology by the RA to support the inspection and to give more insight into animal welfare risks. The lack of external validation for multiple commercially available systems is a point of attention.

## 1. Introduction

### 1.1. Animal Welfare Inspections at the Slaughterhouse

Very high numbers of animals are slaughtered each year. In the Netherlands in 2021 alone more than 500 million broilers, over 17 million pigs and half a million cattle were slaughtered [1]. The slaughter rate is high and can exceed more than 500 pigs, 150 veal calves, 50 cows or 13,000 broilers per hour in large slaughterhouses [2,3]. The high speed and high number of animals make it difficult for humans to assess animal welfare in the slaughterhouse. Furthermore, Regulatory Authorities (RA) and European public media report issues in slaughterhouses with a negative impact on animal welfare several times a year [4,5,6,7]. Despite existing animal welfare legislation and various initiatives by the meat industry and research to reduce animal suffering, animal welfare consequences (i.e., animal welfare problems) still occur on farm, during transport and in the slaughterhouse [8,9,10,11,12,13,14]. 

The slaughterhouse is primarily responsible for the animal welfare at the slaughterhouse and this is monitored by the RA. This monitoring is conducted by visual inspection based on the applicable legislation. European Union (EU) regulation 2017/625 states that inspections on animal health and welfare must be performed at the slaughterhouse, as well as inspections related to meat production such as ante-mortem (AM) and post-mortem (PM) inspection. Historically, food safety has been the main driver for this meat inspection and still is the prime focus at the present [15,16,17,18]. Upon arrival at the slaughterhouse, all animals (individually or at herd/flock level) must receive an AM inspection from an official veterinarian (OV) or under the supervision of the OV (allowed for poultry). The PM inspection is carried out by the OV or an external inspector supervised by the OV [19]. Key components of the AM inspection are related to the handling of the animals, the killing process and the condition of the animals (e.g., clinical signs of disease). The prime focus of the PM inspection is a visual inspection of the carcasses on abnormalities and condemnation. In addition palpation and incisions can be performed and further laboratory analysis can be applied [15,16,17].

The slaughterhouse is a sentinel point in the animal production chain from which animal welfare can be efficiently measured during slaughter as well as retrospectively during earlier phases in the production chain (during transport and on farm) [20,21,22]. Some indicators of animal welfare on farm are easier to measure at the slaughterhouse than on the farm itself. These indicators, for example, only will become visible during a PM inspection, such as stomach lesions or pulmonary disorders in pigs or veal calves or breast blisters in broilers [17,23,24]. Furthermore, the welfare of animals from multiple farms can be assessed at a single location, since animals are brought to the slaughterhouse in large numbers from multiple farms and assembly centres [23,25]. It should however be noted that the number of recorded indicators at the slaughterhouse might be an underestimation of the actual prevalence of the welfare issues on farm level, since on-farm mortality or full recovery after illness are not taken into account [18,26]. 

### 1.2. Use of Animal-Based Measures to Asses Animal Welfare

The welfare consequence experienced by an animal due to a certain situation and the reaction of the animal to the situation, can best be measured directly on the animal itself by using animal-based measures (ABMs) [22,27,28,29]. As stated by Bracke et al.,: “Animal welfare is the quality of life as it is experienced by the animal itself” [30]. Hence ABMs are the preferred approach for measuring animal welfare in a reliable and objective way at a slaughterhouse. Resource- and management-based measures such as type of floor or handling procedures are indirect welfare measures [29]. ABMs may be physiological parameters such as hormone levels, heart rate or blood values; morphometric parameters such as injuries or body weight; or related to behaviour such as vocalisations or movements [31]. An ABM may be the result of one specific event, such as an injury, or the cumulative effect of multiple events over a prolonged time, such as decline in the physical condition of the animal or the development of abnormal behaviour [22,32]. ABMs can therefore provide the RA important information about the preceding production stages. 

However, while ABMs give valuable information on animal welfare, manual scoring of these ABMs is time-consuming and labour-intensive. These properties conflict with the high speed of the slaughter process. Particularly extreme is the case of broilers, where more than 3 chickens per second are presented on the slaughter line for PM inspection [15,33]. Protocols are available for the manual scoring of ABMs at the slaughterhouse, e.g., the Welfare Quality (WQ) protocol for broilers and pigs [34,35]. WQ is an internationally recognised science-based approach for executing a complete animal welfare assessment on farm and at the slaughterhouse. However, the assessment has been shown to be very time consuming; the WQ protocol for the assessment of pig welfare at the slaughterhouse requires an estimated 5.5 to 8 h of human labour per slaughterhouse for a complete assessment of the slaughterhouse [8,20]. But just the scoring of carcass abnormalities alone is already estimated to require one hour for the WQ protocol [35]. As a consequence, unless legally required, manual scoring of additional ABMs will not easily be adopted in the regular meat inspection performed by the OV. Manual scoring is also subject to non-uniformity, inter-observer non-repeatability: different observers can assess significantly different scores [21]. The option of automatic scoring of these ABMs might reduce the need for assessments performed by humans and would alleviate the administrative burden for the RA when additional animal welfare indicators are to be scored.

### 1.3. Sensor Technology and Artificial Intelligence

In recent years, more data on livestock farming has been generated by a number of promising techniques such as sensors e.g., cameras, microphones, speedometers and thermometers [36,37,38,39,40]. The sensor measures an aspect of the environment or of the animal itself. The sensor may be attached to the animal, such as an ear tag, or be placed in the animal’s environment, such as a camera. The data produced can subsequently be processed and analysed using computers and self-learning algorithms. Before we dive into the applications of artificial intelligence (AI) in this context, we would like to give a brief summary of what it entails.

AI refers to the development of computer systems that can perform “intelligent” tasks. It involves creating algorithms and models that enable machines to perceive, reason, learn and make decisions based on input data. Commonly used AI methods include Machine Learning (ML), Deep Learning (DL), Natural Language Processing (NLP), computer vision (CV) and robotics. Machine Learning involves training algorithms with large amounts of data to enable them to learn patterns, make predictions without being explicitly programmed. This can be done in several ways [41]: 

1. Supervised Learning uses labelled data with which the algorithm is trained to learn to “classify” categories;

2. Unsupervised Learning is done with unlabelled data, where the algorithms searches for “similarities” to cluster data together in subsets;

3. Semi-supervised Learning uses a small amount of labelled data with a much larger amount of unlabelled data to train a predictive model.

Deep Learning is a subset of Machine Learning (See Figure 1) that uses artificial neural networks with many layers to process and analyse complex data. This can be supervised, semi-supervised or unsupervised learning. Applications include computer vision, speech recognition, natural language processing, medical image analysis, with results comparable to or surpassing human expert performance [41,42].

There are many examples of sensor and AI use in precision livestock farming (PLF). The main steps of this process are [36,37,38,39,40,42,43]:Collecting data about the health and welfare of the animals from sensors;Labelling of the data by experts or by “smart” sensors;Training the algorithm to classify a wanted or an unwanted event with a certain threshold;Testing the trained algorithm with new data to make sure it works as intended (the more training/testing the better);Using the trained and tested algorithm in real-life situation for flagging certain events;Maintaining/adjusting the algorithm to ensure proper performance.

### 1.4. State of Play on Use of Sensor Technology and AI at the Slaughterhouse and Animal Welfare 

Various (systematic) reviews have been published on the use of PLF on farms to assess animal welfare [37,39,44,45,46,47,48,49,50,51,52,53,54] or on the use of ABMs at the slaughterhouse to manually assess animal welfare [9,11,12,13,23,31,55]. The recent opinions from the European Food Safety Authority (EFSA) highlight the use of ABMs collected in the slaughterhouse to assess on farm welfare and possibilities for automation [18,26,56,57]. Furthermore, a systematic review on the application of computer vision systems at the slaughterhouse has been published by Sandberg et al. [58]. However, the focus of this latter review was on meat safety and not on animal welfare. To the best of our knowledge no recent complete overview exists on the use of sensor technology to automatically assess animal welfare at the slaughterhouse from the viewpoint of an RA. The goal of this narrative review is to provide an overview of the initiatives, opportunities and barriers for regulatory purposes to measure ABMs at the slaughterhouse using sensor technology and AI. The focus of this review is welfare of the most slaughtered animal species in the Netherlands.

The topics in this narrative review are as follows. First, it summarizes current literature on these subjects: (1) ABMs for assessment of welfare in the slaughterhouse for broilers, laying hens, pigs and cattle; (2) the current use of ABMs by the RA; and (3) the use of sensor technology and AI to assess the ABMs. In addition the review lists commercially available systems to measure these ABMs with sensor technology. Finally, it discusses the opportunities and barriers for the use of sensor technology to assess animal welfare at the slaughterhouse in the nearby future and in particular for the RA.

## 2. Methods

### 2.1. Literature Search

The literature search for this narrative review consisted of two parts: (1a) identification of ABMs which can be recorded at the slaughterhouse, (1b) identification of ABMs already part of the current meat inspection and (2) identification of applications where sensor technology and AI were used to measure these ABMs. 

The search for part one was done in September 2022 using PubMed with the terms “welfare” AND (abattoir OR slaughterhouse) AND ((measure) OR (indicator) OR (parameter) OR (outcome)) in all fields. No limitation on publication year. The search string by Brscic et al. [59] was used as an example for selecting the terms. As a second step filters were set on full text available, language in English and review or systematic review as type of document. The filter for (systematic) review was selected out of efficiency reasons, as the goal of this research was to give insight in the opportunities on the use of sensor technology and AI by the RA and not to give an exhaustive overview of all ABMs used in research. The abstract and title of the retrieved papers were manually screened on relevance (related to animal welfare, broilers, laying hens, pigs, veal calves, dairy or beef cows). Relevant literature on ABMs by EFSA and WQ was added in June 2023. The flow chart in Figure 2 describes the search protocol. The EU legislation related to animal welfare during the meat inspection (Regulations (EU) 1/2005, 1099/2009 and 2019/627) [60,61,62] and work instructions [63,64,65,66] for the OV of the Netherlands Food and Product Safety Authority (NVWA) were added to the literature as well. 

For part two PubMed was searched in February 2023 using the terms “welfare” AND ((measure) OR (indicator) OR (parameter) OR (outcome)) AND ((precision livestock farming) OR (sensor)) in all fields. No limitation on publication year. Full text available, language in English and review or systematic review as type of document were set as filters. The filter for (systematic) review was selected out of efficiency reasons, as the goal of this research was to give insight in the opportunities on the use of sensor technology and AI by the RA and not to give an exhaustive overview of forms of sensor technology used in research. Title and abstract of papers were manually screened on relevance (related to animal welfare or animal farming and sensor technology and broilers, laying hens, pigs, veal calves, dairy or beef cows). The flow chart in Figure 3 describes the search protocol. 

### 2.2. Analysis of the Literature

Based on part one of the literature search an overview was made of the ABMs recorded at the slaughterhouse for broilers, laying hens, pigs and cattle, either as an indicator for welfare on the farm, during transport or at the slaughterhouse. Only morphometric and behavioural parameters were taken into account. Physiological parameters or biomarkers were not a part of this review. Similar ABMs were grouped when possible, for instance abnormalities of lungs, heart and other organs were grouped as abnormalities of the organs and different forms of skin damage like lesions, scratches etcetera were grouped as bruises and skin damage. The welfare consequences measured by these ABMs were categorized according to the WQ principles: Good feeding, Good housing, Good health and Appropriate behaviour [67]. The validation of the identified ABM’s was no part of this study.

The relevant legislation and work instructions of the NVWA were scanned on the use of ABMs for animal welfare assessment during the meat inspection at the slaughterhouse in the Netherlands. 

The literature retrieved in steps one and two was analysed on applications where sensor technology and AI were used to measure the identified ABMs. Newly identified review papers in the retrieved literature were scanned for applications as well. For feasibility at the slaughterhouse only applications of technologies that were non-invasive or required no physical handling nor separation of the animals for the assessment were considered in this review. Procedures like attaching a sensor to the animal, placing a sensor inside an animal or separating animals from the group are not practically feasible at the slaughterhouse. As a consequence, technologies like force plates or accelerometers were not considered in this study. A distinction was made between applications with the use of sensor technology and AI, and merely sensors without the use of smart algorithms.

The analysis also contributed to identify commercially available applications of the use of sensor technology and AI at the slaughterhouse and systems currently used by the RA. In addition, 5 experts, scientific researchers, with peer-reviewed publications in the field of sensor technology and animal welfare in the Netherlands were consulted to check for possible missing applications of the use of sensor technology and AI at the slaughterhouse. Furthermore, the webpages of the mentioned European meat processing and slaughter line selling companies identified from the literature and by the experts were visited. Subsequently, applications mentioned during presentations, conferences and webinars attended by the authors were included as well. 

The identified applications were categorized according to the phase of development of the technology: only applied under research settings, in research under practical conditions at the farm or slaughterhouse or commercially available. 

## 3. Recording Animal-Based Measures at the Slaughterhouse

### 3.1. Animal-Based Measures at the Slaughterhouse

ABMs which can be recorded at the slaughterhouse were identified from the 34 retrieved papers in the literature search. For broilers and laying hens 37 ABMs were identified, 32 ABMs for cattle and 41 ABMs for pigs. An overview of the ABMs is presented in Table 1, Table 2 and Table 3. The identified ABMs can roughly be classified into three categories: (1) still image, e.g., pathology or bruising, (2) moving image of the animal, e.g., lameness, falling down or movement after stunning and (3) sounds, such as vocalisations. Examples of retrospective ABMs related to the welfare status on farm or during transport are ABMs based on various pathological abnormalities such as breast blisters, hock burns or abnormalities in the lungs. 

### 3.2. Animal-Based Measures at the Slaughterhouse Recorded by the RA

We examined the relevant work instructions and legislation on the use of ABMs by the Dutch RA at the slaughterhouse. There is a distinction in the procedures at the red meat slaughterhouses (e.g., pig and cattle) and poultry slaughterhouses, especially for the PM inspection.

At a red meat slaughterhouse the OV assesses the fitness for transport of the animals at arrival at the slaughterhouse. Weak, wounded or sick animals are considered not fit for transport. This mainly involves animals that are unable to move painlessly on their own or to walk without assistance, animals with serious open wounds or a prolapse, animals in the last 90% of the gestation period, one week after parturition, or recently born animals without a fully healed navel [62,66]. During the AM inspection the OV determines whether an animal is fit for slaughter. The animal is assessed on the risk of cross contamination during slaughter (animal health and presence of any local clinical abnormalities, cleanliness (dirty animals)), the presence of animal diseases (zoonoses, infectious animal diseases or notifiable animal diseases) and animal welfare. Animal welfare issues alone are not a reason to deny access to slaughter if there is no risk for food safety [61,64]. The work instructions of the NVWA and/or legislation specifically mention some ABMs as attention points such as falling, slipping, vocalisations and signs of consciousness during stunning and the killing process [60,66]. However registration of the observed ABMs during the AM inspection is not legally required and they are not actively registered in the Netherlands. For pigs and cattle the OV only notes down the externally observable pathological abnormality(s) relevant for food safety. This is done on a form in an open text box and passed on for the PM inspection. The way of passing this information on depends on the slaughterhouse location; this can be on paper, by phone or digital system [61,64]. During the PM inspection the carcass and different organs (e.g., heart, liver, lungs, gastrointestinal tract and pleura) are visually inspected on potential risks for human and animal health or animal welfare and abnormalities are noted down [61]. 

In the case of poultry, more ABMs are registered at the slaughterhouse compared to pigs and cattle. In line with EU Regulation 2019/627, the carcasses of broilers are inspected on abnormalities. This is primarily done for food hygiene and safety reasons, but also gives relevant information on animal welfare [26]. During this inspection on flock level carcasses and organs are inspected and scored for five minutes, e.g., on hepatitis and pericarditis. This is done at least once per flock. The results of this inspection are noted down in a digital system (Pladmin) [65]. Footpad lesions are recorded as well. Broiler farmers with a stocking density in category 3 (39 kg/m^2^–42 kg/m^2^) must maintain records of footpad lesion scores as required by the Dutch Animal Keepers Decree (“*Besluit houders van dieren*”) [80]. A certified inspector (per flock, 100 feet per housing unit) or a camera system performs the scoring of the footpad lesions at the slaughterhouse [81]. The requirements of this camera system are imposed in the Animal Keepers Regulation (“*Regeling houders van dieren*”) [82]. The OV also assesses all thinned poultry (earlier removed birds from the flock) on possible poor welfare conditions; two times 50 animals are assessed on footpad lesions and other forms of contact dermatitis (breast blisters and hock burn) [63]. In addition the OV performs some inspections on catching- and transport related injuries on a periodic basis or if there are identified reasons to do so. Examples of these injuries are dark red to purple haemorrhages, sometimes in combination with broken wings or other broken bones. During this inspection the OV counts on at least two occasions to get a clear picture of the whole flock. Each count takes at least two minutes. The percentage of catching-related injuries is calculated by taking the average of the two counts [63]. 

### 3.3. The Use of Sensor Technology and AI to Record Animal-Based Measures

Various technologies are studied in a research setting or under practical circumstances like on farm, during transport or at the slaughterhouse. Several technologies are commercially available. Table 4, Table 5 and Table 6 present an overview of the ABMs that can be recorded at the slaughterhouse using sensor technology (and AI). 

Of the 37 identified ABMs for broilers and laying hens in total 10 applications of sensor technology applied in a research setting, on farm or at the slaughterhouse were reported. Six (16%) were used in research on sensor technology, including a number of studies performed under commercial settings at the slaughterhouse [83,84,85,86,87,88]. For 6 ABMs commercially systems are already available [89,90,91,92,93,94,95,96,97,98]. These systems are designed to address meat quality but can also give valuable information on animal welfare such as prevalence of breast blisters, hock burns or broken wings.
animals-13-03028-t004_Table 4Table 4Development phase for measuring ABMs in broilers and laying hens at the slaughterhouse with sensor technology and AI categorized according to the WQ principles. In brackets are the publications which refer to the combination of Welfare Quality principle, welfare consequence, ABM and system of sensor technology.Broilers and Laying Hens

Development PhaseWelfare Quality PrincipleWelfare ConsequenceABMResearchResearch at SlaughterhouseCommercially AvailableGood feedingProlonged hungerEmaciated animals, body weight

ChickSort 3.0 [89]TrueWeigher 707 [90]SmartWeigher [91]Good healthInjuries (pain)Breast blisters

Meyn [92]ChickenCheck [93]Hock burn

ClassifEYE^®^ [94]ChickSort 3.0 [89]ChickenCheck [93]Footpad lesions
[83,84,85,86]Meyn * [95]ChickenCheck * [93]Plumage damage[99,100,101] ^†^[87]
Wing injuries (bone fractures)

Meyn [92]ClassifEYE^®^ [94]ChickSort 3.0 [89]IRIS [96,97,98]Bruises and skin damage
[88]Meyn [92]ClassifEYE^®^ [94]ChickSort 3.0 [89]IRIS [96,97,98]ChickenCheck [93]DiseaseAscites
[88]
Hepatitis
[88]
Appropriate behaviourFear and painVocalisations[102,103,104,105] ^†^

* are externally validated, ^†^ only sensors, no AI.


Out of the 32 identified ABMs in cattle only technologies for measuring body condition score (BCS), lameness and aggressive behaviour were reported using sensor technology and AI [106,107,108,109,110,111,112,113,114,115,116,117,118,119,120,121,122,123,124,125,126,127,128,129,130,131,132,133,134,135,136,137]. Carcass colour as an ABM for anaemia can be measured with a sensor [138,139], but no automatic system with the use of AI is yet developed. Those four applications represent 12.5% of the identified ABMs.
animals-13-03028-t005_Table 5Table 5Development phase for measuring ABMs in cattle at the slaughterhouse with sensor technology and AI categorized according to the WQ principles. In brackets are the publications which refer to the combination of Welfare Quality principle, welfare consequence, ABM and system of sensor technology.Cattle

Development PhaseWelfare Quality PrincipleWelfare ConsequenceABMResearchResearch at SlaughterhouseCommercially Available SlaughterhouseCommercially Available on FarmGood feedingProlonged hungerBody condition score[106,107,108,109,110,111,112,113,114,115,116,117,118,119,120,121,122,123,124,125,126,140]

DeLaval * [141]4DRT-Alpha [142]Good healthLameness (pain)Lameness[122,127,128,129,130,131,132,133,134,135,136]


AnaemiaCarcass colour
[138,139] ^†^Chromameter ^†^ (e.g., Minolta CR400) [143]
AppropriatebehaviourSocial stressAggressive behaviour[137]


* are externally validated, ^†^ only sensors, no AI.


Of the 41 identified ABMs for pigs, 13 (32%) were reported in applications using sensor technology and AI. Measuring body weight and/or size in pigs using a camera has been studied extensively [144,145,146,147,148,149,150,151,152,153,154,155,156,157,158,159,160,161,162]. Automatic recording of vocalisations as a sign of fear, pain and stress as well [163,164,165,166,167,168,169,170,171,172,173,174,175]. Although, this are studies in an experimental setting and not on automatic recording of vocalisations at the slaughterhouse. Also there are no commercially available systems yet. However, there is a patent for the STREMODO technology [174]. 

Several systems have already been applied in European pig slaughterhouses; systems to record ear- and tail injuries, lung abnormalities, reluctance to move when driving up the pigs, movement after stunning and bleeding rate [176,177,178,179,180,181,182,183]. The commercially available systems of Argus, Genba Solutions and AI4Animals are not only trained on ABMs, but also trained to detect resource- and management-based measures like the noise level, human interaction and the use of prods and stunning devices [179,180,181].
animals-13-03028-t006_Table 6Table 6Development phase for measuring ABMs in pigs at the slaughterhouse with sensor technology and AI categorized according to the WQ principles. In brackets are the publications which refer to the combination of Welfare Quality principle, welfare consequence, ABM and system of sensor technology.Pigs

Development phaseWelfare Quality PrincipleWelfare ConsequenceABMResearchResearch at SlaughterhouseCommercially Available at SlaughterhouseCommercially Available on FarmGood feedingProlonged hungerBody condition[144,145,146,147,148,149,150,151,152,153,154,155,156,157,158,159,160,161,162]

PigWei [184]OptiScan [185] WUGGL [186]GroStat [187] Fancom [188]Good housingCold stressHuddling[189,190,191]


Restricted movementSlipping[192]


Good healthInjuries (pain)Ear injuries
[193]CLK GmbH * [176]
Tail injuries
[193,194,195]CLK GmbH * [176]
Lameness (pain)Lameness[196]


DiseaseAbnormalities organs (lung, stomach, heart, liver, pleura)
[25,197,198]Lung: F4TLaB [177]
Consciousness during stunning and killing processBody movement

Argus [179]Genba Solutions GmbH [180]AI4Animals [181]
Bleeding rate
[199]VisStick * [183]CLK GmbH [182]
Appropriate behaviourSocial stressAggressive behaviour[200,201,202,203,204,205,206] ^†^


Mounting[207,208,209,210,211,212]


FearReluctance to move, freezing[213][192,214]AI4Animals [178]
Fear and painHigh-pitched vocalisations[163,164,165,166,167,168,169,170,171,172,173,174,175]


* are externally validated. ^†^ only sensors, no AI.


### 3.4. Validation

In order to be able to rely on the findings of sensor technology combined with AI, it is important that the AI system is properly validated. There are several statistical methods available to determine the quality of a system. However, to date there are no fixed agreements or accepted methods within the scientific community or determined by EFSA or national RA’s to validate these systems (e.g., no agreements on a sufficient level of accuracy) [36]. There are two forms of validation: external validation, in which the system is evaluated using a completely independent dataset that uses data from animals that was not used in the development of the system, and internal validation, where the system is evaluated using partially the same dataset that was used to build the technology [44,45]. External validation of a system is preferred to prevent overfitting and to handle varying external circumstances [43].

The external validation of the commercially available systems is a point of attention. Of the 20 different commercially available systems only 5 (20%) externally validation studies have been published: the Meyn Footpad Inspection System [83,85], ChickenCheck [84], DeLaval body condition scoring system [107], the system for tail and ear injuries of CLK GmbH [193] and the VisStick system for bleeding rate [199]. Furthermore, the body condition scoring system BodyMat F is validated [104], but this system is no longer commercially available. Although an external validation percentage of 20% for systems in a slaughterhouse setting seems low, it can be considered high compared with the commercially available PLF systems on farm. Gómez et al. [44] found that only 7% of the commercially available and validated PLF systems for pigs were externally validated. In addition, only 14% of the PLF systems for dairy cattle performed external validation [45]. 

## 4. The Use of Sensor Technology and AI to record Animal-Based Measures by the Regulatory Authorities

### 4.1. Existing Technologies and AM and PM Inspection

In general all examples of technologies mentioned in the previous paragraph are relevant for the RA as the OV can use ABMs for an animal welfare assessment at the slaughterhouse. Although there is a distinction between ABMs that serve as attention points for the OV and ABMs that are actively noted down by the OV or inspector. Attention points are for instance lameness, falling or slipping animals, vocalisations and signs of consciousness after stunning. Externally observable pathological abnormalities, organ abnormalities and other issues relevant for food safety are actively noted down. Additionally, in the case of poultry, the number of footpad lesions, breast blisters, hock burn and catching related injuries (bruises and broken wings) are recorded. Table 4, Table 5 and Table 6 show that there are applications of sensor technology and AI for both the ABMs serving as attention points as for the ABMs that are actively noted down. Some systems are already commercially available for application at the slaughterhouse such as systems to record breast blisters, hock burn, footpad lesions, wing injuries, bruises for poultry [89,92,93,94,95,96,97,98] or lung abnormalities, body movement after stunning and bleeding rate for pigs [177,179,180,181,182,183]. Furthermore, ascites and hepatitis for poultry and milk spots in the liver and pericarditis in pigs are studied in a slaughterhouse setting but not yet commercially available [88,197].

VetInSpector (IHFood, Denmark) is an example of potential use of sensor technology and AI by the RA during the PM inspection. This camera- and imaging-analysis system has been developed in Denmark. An existing commercial available system on carcass quality was tested and adapted for the application as part of the PM inspection for broilers. The developed system took pictures of all lesions on the list of the Danish PM inspection, such as hepatitis, ascites and dermatitis. Next, to train the AI model, OVs graded the carcasses on approval for human consumption by using the pictures [88]. The system has been further developed and is accepted in September 2021 as a supporting tool for the PM inspection in Denmark [58].

### 4.2. Current Use of Sensor Technology by the RA

The findings of our study show that certain elements of AM and PM inspections on animal welfare can also be performed by a sensor technology system instead of a human. However, despite the existing commercial systems and development in research, meat is still inspected by humans on the basis of vision, palpation and incision [215,216]. At this moment the Dutch RA uses no sensor and AI technologies apart from video-imaging systems for assessing footpad lesions on broilers at the slaughter line. In Germany and the Netherlands those latter systems have already been applied in some of the larger poultry slaughterhouses [83,84]. Camera surveillance systems (CCTV) are used as well by the RA in several countries, among others in the Netherlands and the United Kingdom. Processes with live animals at the slaughterhouse are recorded, the camera footage can be reviewed by the RA on site and used to support enforcement [217,218].

### 4.3. Legal Framework and Use of Sensor Technology by the RA

EU Regulation 2019/627 states that official controls must be either performed by an OV or take place under the supervision of the OV. For this reason an assessment solely by sensor technology is not accepted as official control and the deviations established by the system should be confirmed by an OV. However, there are some exceptions in the regulation. According to EU Regulation 2019/627 only a representative sample of poultry from each flock must receive a PM inspection if: “food business operators have a system in place to the satisfaction of the official veterinarian, that allows the detection and the separation of birds with abnormalities, contamination or defects;”. Although criteria for a satisfactory system (e.g., required level of accuracy) are not set in the legislation. An example of such a system is the video-imaging system for the assessment of footpad lesions on broilers at the slaughter line applied in some German and Dutch slaughterhouses [83,84]. Furthermore, on the long term there are opportunities for other inspections or animal species as well. EU Regulations 2017/625 and 2019/627 give room for scientific and technological development. However, this procedure is more complex as approval by the European Commission and other European Member states is necessary: “For the purpose of developing new control methods and techniques in relation to official controls on meat production, competent authorities should be allowed to adopt national measures to implement pilot projects that are limited in time and scope. Such measures should ensure that competent authorities verify that operators comply with all the fundamental provisions applicable to meat production, including the requirement that meat is safe and fit for human consumption. In order to ensure that the Commission and the Member States have the possibility to assess the impact of such national measures and express their opinion before they are adopted, and take therefore the most appropriate action, those measures should be notified to the Commission [..]” and “The Member States shall inform the Commission and other Member States on scientific and technological developments, [..] for consideration and further action as appropriate”.

## 5. Discussion

To the best of our knowledge, the current study is the first to provide an overview on the use of sensor technology to automatically assess animal welfare indicators at the slaughterhouse from the viewpoint of an RA. Many ABMs for broilers, laying hens, pigs and cattle can be recorded at the slaughterhouse. Some of these ABMs can be measured with the use of sensor technology and AI. This provides opportunities for the use of several of these technologies as part of the meat inspection by the RA.

### 5.1. Research on Sensor Technology to Record Animal-Based Measures

An interesting finding of our study is the notable discrepancy in the number of identified ABMs at the slaughterhouse and applications reported on the use of sensor technology and AI to record these ABMs in a research setting, on farm or at the slaughterhouse. Applications of sensor technology were reported for 10 of the 37 ABMs (27%) identified for poultry, 4 of 32 for cattle (12.5%) and 13 of 41 (32%) for pigs. These results are in line with the findings of Sandberg and colleagues [58]. They found very few publications in their systematic review on the use of camera-vision systems for meat safety assurance. Possible reasons may be the constraints by legislation and insufficient return on investment [58].

Although relatively few ABMs have been reported in the examples using sensor technology and AI, numerous studies have explored the use of sensor technology and AI to assess these ABMs in a research or farm setting. However, limited research is available on the use of these technologies at the slaughterhouse. An example is the automatic recording of body condition score and weight of pigs. This is extensively studied in research settings and on farm [144,145,146,147,148,149,150,151,152,153,154,155,156,157,158,159,160,161,162] and commercial systems for use on farm are available as well [184,185,186,187,188]. Non-uniformity of pig size and weight in a batch from a farm at the slaughterhouse is a relevant indicator of feeding related welfare issues on farm. Maisano et al., showed that pigs that weighed more than 30 kg less than the Italian standard for heavy pigs of 160–170 kg were shown to have had poor growth [28]. However, despite the relevance of recording this ABM at the slaughterhouse, no research was conducted at the slaughterhouse yet. Similar findings apply to cattle; automatic scoring of body condition is extensively studied in research settings on farm [106,107,108,109,110,111,112,113,114,115,116,117,118,119,120,121,122,123,124,125,126,140], multiple systems are commercially available to automatically assess BCS on farm [141,142,219], but research at the slaughterhouse is missing. Another relevant on farm commercial available technology might be the coughing detection system Soundtalks [220]. The detection of coughing pigs at the slaughterhouse might be a useful tool for detecting sick animals. 

Further research is needed to validate on farm systems for their use in a slaughterhouse, as the situation differs from the farm setting. In the example of the body condition scoring system for cattle, the system is integrated in the milking system. The cow individually enters the milking parlour and stands still. In contrast to the moving cows in groups at the slaughterhouse. Grouping of animals is a familiar problem in the application of sensor technology under a commercial setting. Attempts to implement a computer vision system for assessing lameness on a commercial dairy farm brought up some issues and the images could not be analysed. In the experimental setting, the animals walked in front of the camera calmly one by one. While at the dairy farm the main problems were the rapid flow of animals, not capturing only one cow per image and cows stopping or running [127,133]. 

### 5.2. Observations by a System Versus Human Observation 

Implementing the use of a system with sensor technology and AI in the work of the OV could have a significant impact on the inspection work by the OV. It is important to gain insight in the impact of this implementation by comparing observations by the OV with observations with sensor technology and AI. However, scientific studies comparing the applied systems at the slaughterhouse with real time observations at the slaughterhouse are limited. Van Harn and de Jong mentioned that it was not possible to make this comparison due to the speed of the slaughter line [83]. Often a comparison is made between pictures scored by the video-imaging system and pictures scored by a trained observer [25,83,84,86,87,88,194,195,197,198]. The scoring based on a picture by an observer is not equal to scoring in a real time slaughterhouse setting. In the development of the TailCam a comparison was made on a small scale between on the real time observation of tail lesions at the slaughterhouse and visual observation based on pictures. For 188 out of 218 tails (86%) there was agreement between the assessment based on pictures and at the slaughterhouse [195]. Blömke et al., compared the agreement between the system and human observation both based on pictures as on real-time observations at conveyor belt at the slaughterhouse. The agreement (Krippendorff’s alpha (α)) between the skilled veterinary observer based on pictures and on direct assessment at the slaughterhouse was 0.89 for ear lesions and 0.71 for tail lesions respectively [193]. Those two studies might indicate that a comparison between picture based scoring and the camera-based system seems acceptable, but more research is needed to confirm this. 

Both Jung et al., and Blömke et al., compared the observations of a camera-based system with human observations at the slaughterhouse. The prevalence-adjusted, bias-adjusted Kappa (PABAK) of automatic assessment of keel bone damage in laying hens compared to human assessment at the slaughter line was 0.72, indicating a substantial reliability [87]. However the slaughter line speed did not exceed one carcass per 2 s, while under commercial practices this can be 0.4 s per carcass [33,87]. The agreement (Krippendorff’s alpha (α)) between the skilled veterinary observer based on direct assessment at the slaughterhouse was 0.62 for ear lesions and 0.55 for tail lesions compared to the camera-based system. This is lower as the α between the skilled veterinary observer based on pictures and the system, 0.64 for ear lesions and 0.75 for tail lesions, respectively. The authors suggest that these lower values on the agreement for the direct assessment might be the result of the limited time to assess the carcass at the conveyor belt (8 s) and the position of the carcass compared to the observer; the ears of the carcass are easier to observe as the tail [193]. In both studies the prevalence scored by the camera-based system differed from the prevalence score by the real time human observations. In the research on tail- and ear lesions the prevalence based on the system was higher, while in the research on keel bone damage the prevalence scored by the system was lower [87,193]. This is an important factor to take into account when these scores are used for enforcement by the RA. Therefore, not only the comparison of scoring based on pictures and the system should be part of the validation study, but a comparison with real time observations by a human observer as well. 

### 5.3. Commercial Application 

A second discrepancy found as a result of our study is the discrepancy between applications in research and commercially available technologies. For many of the commercial available systems there is no scientific research directly related to the applied system. For example, the technologies to detect carcass abnormalities in poultry or body movement after stunning in pigs. These results are also in line with the findings of Sandberg and colleagues. They argue that companies might keep their knowledge confidential until the system is fully implemented and validated [58]. In addition most of the commercially available systems at the slaughterhouse were designed for meat quality and not assessment of animal welfare or official controls. The development and implementation of these systems is economically driven and not hindered by legislation. The same applies to the implementation of PLF systems on farm. Return of investment is an important driver. The most successful commercially available systems are related to increased production and convenience for the farmer, not animal welfare. For instance the system on estrus detection in cattle [43]. 

### 5.4. Legal Barriers

Currently there are limited options for the RA to use sensor technology and AI as part of the meat inspection at the slaughterhouse. The present EU legislation is the most prominent barrier as official controls must be either performed by an OV or take place under the supervision of the OV. As a result sensor technology cannot yet (fully) legally replace the OV. Our results regarding animal welfare are in line with the review studies of Nagel-Alne et al., and Blagojevic et al., on meat safety legislation and the constraints for innovation; the multiple normative demands (“how it should be done”) instead of the functional demand (“what needs to be achieved”) of the EU legislation on meat safety may hinder the development of innovative systems [221,222]. There are some exceptions for the post mortem inspection of poultry as a system to detect and separate birds with abnormalities is allowed to replace a full PM inspection by a representative sample if the system is approved by the OV. For other animal species a new control system needs to be approved by the EU Commission and member states. This legislation should be adapted to the same legislation as applicable to poultry to simplify this procedure and to stimulate innovation [58]. 

A second legal barrier are the many qualitative goal-oriented standards, also referred to as open standards, in the animal welfare legislation [223]. When developing sensor technologies combined with AI, thresholds must be established in order to be able to create a classification with a target value. In the case of open standards, generally accepted target values have often not been established and enforcement cannot take place solely based on an observed value. For example, there is no legal standard on the maximum number of vocalisations as sign of pain or distress and when this number of vocalisations reaches the threshold of avoidable form of pain, distress or suffering. Similarly, there are no legal standards for the maximum number of cases of pneumonia in pigs. The OV and/or inspector must provide reasons substantiating why in this specific situation, based on the established facts, the open standard was violated. 

The quantitative goal-oriented standards in animal welfare legislation offer more opportunities for the use of sensor technology and AI by the RA. Since a closed standard such as a limit value is set out in the law. The previous mentioned video-imaging systems for assessing footpad lesions on broilers at the slaughter line is for instance related to a defined score in the Dutch Animal Keepers Decree. Another example is the standard that animals may only be killed when they are unconscious and insensitive (with the exception of ritual slaughter) (Council Regulation (EC) No 1099/2009). No signs of awareness, consciousness or sensitivity must be present between the end of the stunning process and the animal’s death. Relevant technologies for enforcement of this standard are the various systems that detect body movement or measure bleeding rate in stunned pigs [179,180,181,182,183]. 

### 5.5. Opportunities on the Use of Sensortechonology and AI by the RA

Despite the legal limitations sensor technology and AI can already contribute to the animal welfare inspections of the RA at the slaughterhouse. The retrieved data from the systems at the slaughterhouse can give the RA some valuable (extra) information. Firstly, the data on the prevalence of the welfare consequences and exposure to hazards can be used for the purpose of risk assessment and help to improve and facilitate the supervision and inspections by the RA. Data from the already commercially available and present systems in the slaughterhouses could be used (anonymously) to represent the prevalence of various welfare consequences such as broken wings, bruises and skin lesions in poultry. This data can not only provide the RA with relevant information about welfare issues at the slaughterhouse, but also valuable information about the welfare on farm. Secondly, the data can help the OV with enforcement by substantiating a welfare issue and deviation or in the identifying of farms with welfare issues for the risk-based inspections. As a third option, slaughterhouses can develop and implement systems as part of a quality system. The owner of the quality system must itself submit the application for acceptance to the RA. The RA will then test whether the quality system satisfies the conditions and criteria and the system safeguards animal welfare [224,225]. Summarizing these three options: systems of sensor technology and AI can be part of an integrated risk-based meat safety assurance system (RB-MSAS) in the future [58,222]. 

A prerequisite for the use of this data is access to the various data sources by the RA. However, at this moment the data obtained by the applied systems is owned by the slaughterhouse and not freely available for the RA. As a result the RA is dependent on the collaboration of the slaughterhouse for use of the data and issues on data ownership and privacy play a role. An example is the registration of carcass colour in veal calves. Carcass colour is an ABM for anaemia. This data is registered at the slaughterhouse, but not directly accessible for a welfare assessment by the OV [18]. 

Besides the collection of data, there are more advantages for the RA for the use of sensor technology and AI. At first, cameras are capable of capturing images at high speed, which can be analysed at a later stage. Due to the high processing speed at the slaughterhouse, it is difficult for assessors to visually inspect individual animals: for example, for poultry less than one second is available to identify abnormalities in each carcass [15,33,226,227]. Theoretically all events at the slaughter process can be recorded 24/7 using sensor technology, which can subsequently provide more information on the welfare of animals following analysis compared to intermittent supervision by a human inspector. Using sensors and AI, data can be captured in a continuous and standardised manner compared to manual scoring by observers, which is very time consuming and may also lead to variation in scores between observers [25,58]. Especially borderline cases can lead to different scores and conclusions [58]. Sandberg and colleagues observed for instance 30% disagreement between the OV’s in grading and rating of pictures from poultry carcasses on lesion severity and acceptance for human consumption in the development of the VetInspector system (IHFood, Denmark) is [88]. An automatic system allows for more uniform scoring [197]. Secondly, no contact with animals or carcass is required. A camera or a microphone can easily be installed within a space and will not cause any additional distraction, fear or stress to the animal, in contrast to presence of humans [214,227]. Due to the fact that any cameras or microphones would be installed within a given space and would not be connected to an individual animal, this allows for assessment of the welfare of several animals at once instead of only one animal at a time [51,52]. Automatic measurement of indicators does not require direct contact with the animals and likewise allows the welfare of smaller animals, such as poultry, to be measured [38,228]. The various technologies also have potential in the area of PM inspections by reducing the risk of cross-contamination [197].

Despite these numerous advantages, sensor technology combined with AI is insufficient to fully replace humans at present. Many systems that have been developed only measure a single type of anomaly, i.e., the one for which the system has been trained. Several technological systems are required in order to detect multiple aspects. Humans are able to detect and identify multiple anomalies at the same time and will also identify less common welfare consequences [15,43,229]. In a similar manner is one system of sensor technology not able not perform a full welfare assessment and to provide information on the actual welfare status of the animal; the animal welfare status is a combination of many factors, not the absence of one welfare consequence.

### 5.6. Limitations of This Study

This narrative review gives an extended overview of studies on the possible use of sensor technology and AI to assess ABMs at the slaughterhouse. Nonetheless, recently published relevant papers might be missing, as our study is not a systematic review. The findings of this review paper are primarily based on other systematic review papers. However, the goal of this research was to give insight in the opportunities on the use of sensor technology and AI by the RA and not to give an exhaustive overview of all technologies currently available. In this light, the authors feel this approach was justifiable for the research goal. 

Another source of uncertainty is the validation of the identified ABMs. A prerequisite of a reliable system of sensor technology and AI is not merely a validated algorithm, but firstly also a validated ABM [43,84]. The ABM used as gold standard must be able to identify a specific welfare consequence and must be repeatable and reliable [22,32,230,231]. Validation of the identified ABMs was not the focus of this study, but is an very important point when developing a system and in validation of the system for use by the RA. Nevertheless, we used scientific literature to identify the ABMs and most ABMs are identified in multiple literature sources. We used these identified ABMs as input for the search on applications of sensor technology. With this approach we expect the used ABMs in the applications of sensor technology to be reliable ABMs. 

### 5.7. Future Research

Recent years have seen many developments in the use of sensor technology and AI to measure animal welfare. However, the current study identified several research gaps: there is no research on the use of sensor technology and AI by the RA at the slaughterhouse, no research on the use of sensor technology and AI to assess ABMs for cattle at the slaughterhouse and validation studies lack for many of the commercial available systems. 

The various studies and commercial applications among the different animal categories show the potential of these applications for other animal categories as well. For instance research on detection of carcass abnormalities is available for pigs and poultry [25,88,197,198] and commercial systems are available as well [89,92,93,94,96,97,98,177]. These applications could be translated for use in cattle slaughterhouses. Also commercial systems of detection of movement or bleeding after stunning in pigs [179,180,181,182,183] raise opportunities for other animal species.

Before the RA can use sensor technology some hurdles need to be overcome: (1) systems should be developed to assess ABMs at the slaughterhouse, (2) these systems should be externally validated, (3) comparison to human observation at the slaughterhouse should be part of the validation, (4) guidelines or agreements on criteria for validation should be formed by the RA (when is a system “satisfactory”?) (5) the data should be available for the RA and (6) the RA should legally be allowed to use these systems. The EU legislation gives room for pilot studies, which can contribute to updating the legislation [222]. To achieve this more research at the slaughterhouse is needed with special focus on the development of systems at the slaughterhouse for cattle, external validation of developed systems (including comparison to human observations) and the use of sensor technology and AI for assessing animal welfare as part of meat inspection. Setting guidelines for validation and giving incentives for the development of these systems at an European level could help to overcome these hurdles. 

## 6. Conclusions

The main goal of the current study was to provide an overview of the initiatives, opportunities and barriers for regulatory purposes to measure ABMs at the slaughterhouse using sensor technology and AI. This study has identified ABMs for broilers, laying hens, pigs and cattle that can be recorded at the slaughterhouse. An ABM measured at the slaughterhouse can provide valuable information regarding the welfare of the animal at the time of slaughter as well retrospectively during its stay on farm and during transport. Part of these ABMs can measured with the use of sensor technology and AI as is shown in studies in a research or farm setting. In contrast, limited research is available on the use of these technologies at the slaughterhouse, but several commercial systems are available to record ABMs at the slaughterhouse. 

On the level of individual animals, sensor technology combined with AI mostly gives opportunities for measurements carried out on the carcass such as detection of abnormalities. Regarding ABMs of live animals at the slaughterhouse the opportunities lie primarily in monitoring groups of animals. For instance, behavioural abnormalities as aggressive behaviour in the waiting pen or slipping or freezing animals during unloading. These observations can be combined with audio analyses to detect vocalisations as signs of stress. 

The findings of our study show that aspects of AM and PM inspections on animal welfare can also be performed by a system of sensor technology in cases such as abnormalities on carcasses like broken wings, bruises or pericarditis. However, most systems are only trained to measure one anomaly and legal barriers exist because EU Regulation states that the AM and PM inspection must be performed by an OV or under the supervision of the OV. Also external validation of available systems and comparison with human observations at the slaughterhouse are a points of attention. Therefore at this moment sensor technology and AI cannot yet fully replace a human to assess animal welfare at the slaughterhouse. Although it does provide many opportunities for the RA to support a welfare assessment and gives more insight in existing animal welfare risks. Therefore systems of sensor technology and AI at the slaughterhouse can improve inspections and supervision performed by the RA in the future [232].

## Figures and Tables

**Figure 1 animals-13-03028-f001:**
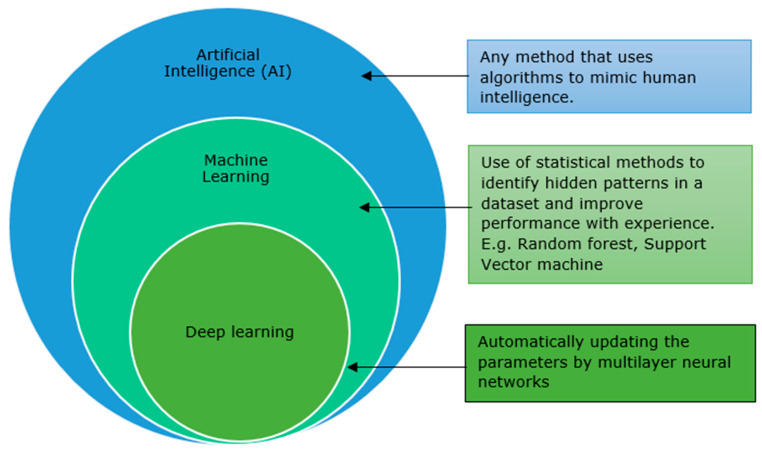
Schematic representation of deep learning and machine learning as part of artificial intelligence. Modified from Shimizu, H. and K. I. Nakayama [42].

**Figure 2 animals-13-03028-f002:**
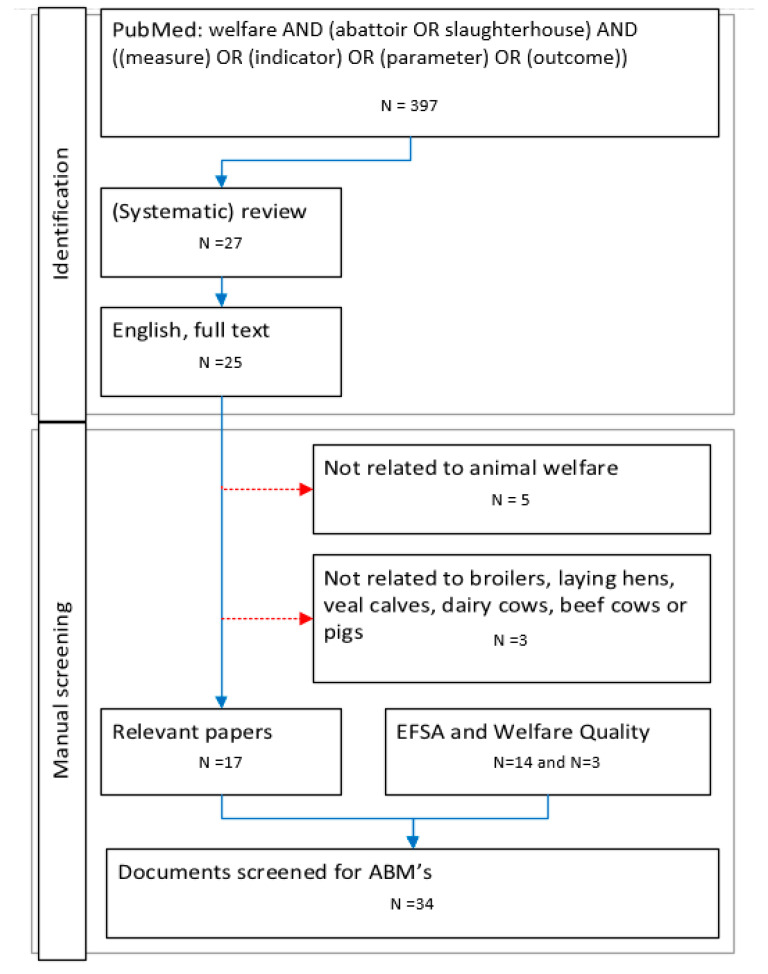
The flow chart represents the steps in the literature search on ABMs at the slaughterhouse. Dotted red arrows pointing to the right represent papers excluded during the manual screening.

**Figure 3 animals-13-03028-f003:**
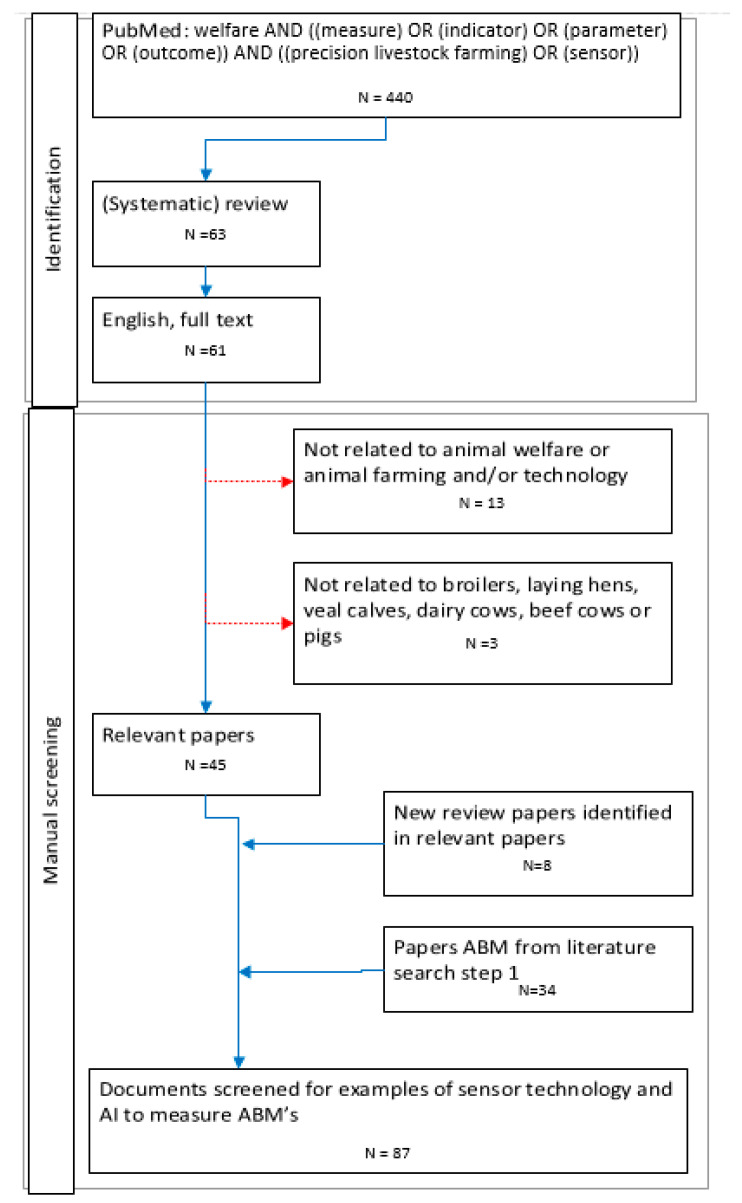
The flow chart represents the steps in literature search on applications of sensor technology and AI to measure ABMs at the slaughterhouse. Dotted red arrows pointing to the right represent papers excluded during the manual screening.

**Table 1 animals-13-03028-t001:** The ABMs identified in literature and assessed at the slaughterhouse in broilers and laying hens are categorised according to the WQ principles. The ABMS are assigned to the scenarios to which the welfare assessments carried out at the slaughterhouse relate (farm, transport and slaughterhouse). In brackets are the publications which refer to the combination of Welfare Quality principle, welfare consequence, ABM and relevant scenario.

Broilers and Laying Hens			Relevant Scenarios for Welfare Assessment at the Slaughterhouse		
Welfare Quality Principle	Welfare Consequence	ABM	Farm	Transport	Slaughterhouse
Good feeding	Prolonged hunger	Presence bile/urates/orange discharge at bottom containers		[11]	[11]
		Emaciated animals, body weight	[26,34,56,68]		
	Dehydration	Dehydrated animals	[34,68]		
Good housing	Heat stress	Panting		[11]	[11]
	Cold stress	Huddling		[11]	[11]
		Piloerection		[11]	[11]
		Shivering		[11]	[11]
	Limited movement	Pilling up (overcrowding in container)		[11]	[11]
	Proper housing	Dirty animals	[69]		
Good health	Injuries (pain)	Breast blisters	[26,34,68,69]		
		Hock burn	[34,68,69]		
		Footpad lesions	[26,34,56,68,69]		
		Plumage damage	[26]		
		Keel bone fracture	[26]	[26]	[26]
		Wing injuries (bone fractures)		[11,68,69]	
		Bruises and skin damage	[56,68]	[11,56,68,69]	[11,56,68,69]
	Disease	Ascites	[26,34,68]		
		Arthritis	[26,56]		
		Septicaemia	[26,34,56,68]		
		Hepatitis	[26,34,56,68]		
		Pericarditis	[26,34,56,68]		
		Abscesses	[26,34,56]		
		Parasites	[68]		
	Death	Dead on arrival (DOA)		[11,69]	
	Respiratory stress during stunning	Deep breathing			[11]
		Hyperventilation			[11]
	Pain during stunning/killing	Muscle spasms			[11,70]
		Withdrawal reflex			[11]
	Consciousness during killing process	Eye blinking			[69]
		Corneal reflex			[70]
		Attempt to regain posture			[11]
		Maintenance of posture			[11]
Appropriate behaviour	Fear	Bunching			[11]
		Wing flapping			[11,69,70]
	Fear and pain	Escape attempts			[11]
		Head shaking			[11,69]
		Vocalisations			[11,69]

**Table 2 animals-13-03028-t002:** The ABMs identified in literature and assessed at the slaughterhouse in cattle are categorised according to the WQ principles. The ABMS are assigned to the scenarios to which the welfare assessments carried out at the slaughterhouse relate (farm, transport and slaughterhouse). In brackets are the publications which refer to the combination of Welfare Quality principle, welfare consequence, ABM and relevant scenario.

Cattle			Relevant Scenarios for Welfare Assessment at the Slaughterhouse		
Welfare Quality Principle	Welfare Consequence	ABM	Farm	Transport	Slaughterhouse
Good feeding	Prolonged hunger	Body condition score	[18,31,69,71]		
	Prolonged thirst	Increased aggression at drinking trough		[13]	[13]
Good housing	Heat stress	Panting			[13]
	Cold stress	Shivering			[13]
	Restricted movement	Slipping			[13,27,31,72,73,74]
		Falling			[13,27,31,69,72,74]
	Comfortable resting	Swollen hocks or bursa	[18,69]		
		Dirty animals	[69]		
Good health	Injuries (pain)	Bruises and skin damage		[13,27,69,72]	
		Broken bones		[13]	
	Lameness (pain)	Lameness	[71]	[13,27,69,72]	[13,69,72,73]
	Anaemia	Carcass colour	[18]		
	Disease	Abnormalities organs (lung, rumen, heart, liver, intestine, udder)	[18,31,75]		
	Fatigue	Exhaustion		[13,69]	[13,73]
		Rapid breathing (tachypnoea)		[13]	
	Consciousness during stunning and killing process	Remaining posture			[13,27,31,69,70,72,73,76,77,78]
		Body movement			[13,31,70,72,73,76]
		Breathing			[13,27,31,69,70,72,73,76,77]
		Tonic and clonic seizure			[13,27,78]
		Cornea and/or palpebral reflex			[13,27,31,70,72,73,76]
		Spontaneous blinking			[13,27,31,69,70,72,73,76,77]
		Eye movements			[13,27,31,70,72,73,77]
		Muscle tone			[13,31]
		Response to nose prick			[77]
Appropriate behaviour	Social stress	Aggressive behaviour			[13,31]
		Mounting			[13]
	Fear	Escape attempts			[13,31,78]
		Turning around or moving backwards			[13,27,31,72,78]
		Struggling in the stunning box (kicking)			[27,31,72]
		Jumping in the stunning box			[27,31,72]
	Fear and pain	Reluctance to move, freezing			[13,27,31,72,78]
		Vocalisations			[13,27,31,69,70,72,73,74,76,77,78]

**Table 3 animals-13-03028-t003:** The ABMs identified in literature and assessed at the slaughterhouse in pigs are categorised according to the WQ principles. The ABMS are assigned to the scenarios to which the welfare assessments carried out at the slaughterhouse relate (farm, transport and slaughterhouse). In brackets are the publications which refer to the combination of Welfare Quality principle, welfare consequence, ABM and relevant scenario.

Pigs			Relevant Scenarios for Welfare Assessment at the Slaughterhouse		
Welfare Quality Principle	Welfare Consequence	ABM	Farm	Transport	Slaughterhouse
Good feeding	Prolonged hunger	Body condition	[57]		
	Prolonged thirst	Increased aggression at drinking trough		[12]	[12]
Good housing	Heat stress	Panting		[9,12,27,35]	[9,12,27,35,55]
		Discolouration of the skin		[12]	[12]
	Cold stress	Shivering		[12,27,35]	[12,27,35]
		Huddling		[9,12,27,35]	[9,12,27,35]
	Restricted movement	Slipping			[9,12,27,35,55]
		Falling			[9,12,27,35,55,69]
	Proper housing	Bursitis	[23,57]		
		Dirty animals	[69]		
Good health	Injuries (pain)	Bruises and skin damage	[23,27,57,69]	[12,23,27,32,35,69]	[9,12,27,32,35,69,79]
		Broken bones		[12]	[12]
		Ear injuries	[32,57]		
		Tail injuries	[23,32,57]		
		Vulva lesions	[57]		
	Lameness (pain)	Lameness	[57]	[12,27,35,57,69]	[12,27,35,57,69]
	Fatigue	Exhaustion		[9,12,69]	[9,12]
		Shortness of breath and open mouth breathing (dyspnoea)		[12]	[12]
		Muscle tremors		[12]	[12]
	Disease	Abnormalities organs (lung, stomach, heart, liver, pleura)	[23,27,32,35,57]		
		Abscesses	[57]		
		Sick animals		[27,35]	[27,35]
	Death	Dead animals		[9,27,35,69]	[9,27,35]
	Respiratory stress during CO_2_ stunning	Gasping for air			[12,70,76]
		Hyperventilation			[12]
		Head shaking			[12]
	Consciousness during stunning and killing process	Remaining posture			[12,27,35,55,69,70,76]
		Body movement			[12,70,76]
		Breathing			[9,12,27,35,55,69,70,76]
		Tonic and clonic seizures (muscle tone)			[12,27,70]
		Cornea and/or palpebral reflex			[9,12,35,55,70,76]
		Spontaneous blinking			[9,12,27,69,70,76]
		Eye movements			[12,70]
		Response to nose prick or ear pinch			[12,55,69]
		Bleeding rate			[55]
Appropriate behaviour	Social stress	Aggressive behaviour			[12]
		Mounting			[12]
	Fear	Reluctance to move, freezing			[9,12,27,35]
		Turning or walking backwards			[9,12,27,35]
	Fear and pain	High-pitched vocalisations			[9,12,27,35,69,70,76]
		Escape attempts			[12]

## Data Availability

No new data were created or analyzed in this study. Data sharing is not applicable to this article.

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
