# Peer review of "Opportunities for Regulatory Authorities to Assess Animal-Based Measures at the Slaughterhouse Using Sensor Technology and Artificial Intelligence: A Review"

_animals, 2023, doi:10.3390/ani13193028_

Round 1
Reviewer 1 Report
Overall
Overall, this is a relevant, comprehensive and well performed narrative review.
Abstract
Line 15-16, it is stated that sensor technology has the potential to replace humans. In discussion line 579, it is phrased that it can give “some valuable (extra) information” so maybe the abstract should to a little higher degree reflect the limitations mentioned in the paper
Introduction
Line 44: Check if “incident” should be “observation”
Line 64-66 states: “The meat inspection at the slaughterhouse is a sentinel point in the animal production chain from which animal welfare can be efficiently measured during slaughter as well as retrospectively during earlier phases in the production chain”.
Here I find the word “efficiently” to be quite a postulate as there can be huge differences between meat inspectors. And in line 75-77, the authors also indicate that the recorded indicators may be an underestimation of welfare and in line 615-616 differences between OVs are mentioned. Therefore, I suggest the expression “efficiently” to be stated more careful. And it could be more clear that there are two levels of comparison: 1) sensor data compared to observations at slaughter house and 2) Relation between sensor data/observations and the actual animal welfare. This distinction could be more clear. It seems that the paper mostly addresses number 1 – and if this is made clear, it could justify a statement like “more efficiently”
Later I saw Line 650 that: “Validation of the identified ABM’s was not the focus of this study” – this information would be relevant also to include in the introduction.
Line 167-168: Here it is stated that the focus is on welfare. I find that a focus on also animal health would be relevant. Can you give some argument why ‘only’ welfare is in focus.
Line 186: The term “meat inspection” could also have been included as search term. Do the authors have some thoughts if this could have retrieved more publications?
Line 189: Why are only review or systematic review included as type of document? Original research publications could also add relevant information – some explanation is given later in line 640-643, but arguments could also be given here.
Line 245: Suggest to elaborate a bit on the competences of the 5 mentioned experts
Line 232-254: The word “example” is used many times – how should this be understood? Did you only choose some of these “examples” that you found?
Recording animal-based measures at the slaughterhouse
Table 1-3: Horisontal lines between the 4 welfare quality principles would ease the overview for the reader.
Section 2.3 line 283: Suggest to mentioned the total number of ABM’s (also mentioned in abstract) for each species here – or maybe give them numbers in Table 1-3 – this makes it easier for the reader to keep track of differences.
Line 406-431: In the description of the principles of technologies only a few details is given, e.g. camera, but it would be nice if the technology behind could be elaborated a bit more.
Line 469 and 471: Not sure what is meant by the two parentheses: (..)
Discussion
Nice and thorough discussion.
The previous mentioned two levels of comparison could be elaborated more in discussion, i.e.: 1) sensor data compared to observations at slaughter house and 2) Relation between sensor data/observations and the actual animal welfare.
Author Response
Dear reviewer,
Thank you very much for taking the time to review this manuscript. Please find the detailed responses below and the corresponding revisions/corrections in track changes in the attachment.
The comments of the reviewer are in bold.
Kind regards,
Abstract
Line 15-16, it is stated that sensor technology has the potential to replace humans. In discussion line 579, it is phrased that it can give “some valuable (extra) information” so maybe the abstract should to a little higher degree reflect the limitations mentioned in the paper
Agree, because of the limitations it will probably not fully replace a human. The text is rephrased ‘to play an important role. See L 17
Line 44: Check if “incident” should be “observation”
Adjusted to issues, see L45
Line 64-66 states: “The meat inspection at the slaughterhouse is a sentinel point in the animal production chain from which animal welfare can be efficiently measured during slaughter as well as retrospectively during earlier phases in the production chain”.
Here I find the word “efficiently” to be quite a postulate as there can be huge differences between meat inspectors. And in line 75-77, the authors also indicate that the recorded indicators may be an underestimation of welfare and in line 615-616 differences between OVs are mentioned. Therefore, I suggest the expression “efficiently” to be stated more careful.
Efficiently refers to the place of the inspection: the slaughterhouse. Sentence is changed, also accordingly to reviewer 2. See L68
An extra paragraph (5.2.) is added in the discussion on the comparison of sensor data and human observations at the slaughterhouse.
The fact that the validation of the identified ABM’s was no part of this study is added to the methods section. See L239
Line 167-168: Here it is stated that the focus is on welfare. I find that a focus on also animal health would be relevant. Can you give some argument why ‘only’ welfare is in focus.
We included animal health as part of animal welfare. Diseases are part of the Good health principle.
Parts of animal health solely or mostly related to food safety (e.g. salmonella) are not part of the scope of this review. Commercial applications of sensor technology at the slaughterhouse have to date generally been focused on food safety and product quality, and far less on animal welfare. Therefore we decided to focus on animal welfare. Also to make the focus of this research manageable.
Line 186: The term “meat inspection” could also have been included as search term. Do the authors have some thoughts if this could have retrieved more publications?
We agree with the reviewer that adding meat inspection to the search string: AND (abattoir OR slaughterhouse OR meat inspection) could have given more results. However, we did not include meat inspection as a search term, because the starting point was animal welfare in general at the slaughterhouse and not only as part of the meat inspection. Animal welfare is only a small part of the meat inspection as the focus is primarily on food safety.
To check if the addition of meat inspection would have given more publications we did a new search in September 2023. The search string with “welfare” AND (meat inspection) AND ((measure) OR (indicator) OR (parameter) OR (outcome)) and applied filters review, systematic review, and English only gives 6 results. Only 2 are related to animal welfare and both are also found with the original search string.
Line 189: Why are only review or systematic review included as type of document? Original research publications could also add relevant information – some explanation is given later in line 640-643, but arguments could also be given here.
We agree with the reviewer that the addition of original research publications in the search string could also have given more relevant information. However as indeed mentioned in line 640-643 we limited our search to (systematic) review papers, because the goal was not to be give an exhaustive overview, but on giving insight in the opportunities. Not limiting to (systematic) review papers would have been very time-consuming and presumably not give different insights. We took into consideration that there are already quite some (systematic) review papers on animal-based measures or on precision livestock farming. We therefore chose to elaborate on the work of other researchers, instead of performing a similar research.
This consideration is added now added in the methods section. See L197-200 and L218-221
Line 245: Suggest to elaborate a bit on the competences of the 5 mentioned experts
Scientific researchers is added between brackets, see L260
Line 232-254: The word “example” is used many times – how should this be understood? Did you only choose some of these “examples” that you found?
We agree with the reviewer that by using the word example this implicates that we only chose some of these examples. This however was not the case. All mentioned technologies were listed and together they are examples of the use of sensor technology to measure animal welfare. Using the term examples is thereby confusing. “Example” is therefore replaced with “application” when relevant.
Recording animal-based measures at the slaughterhouse
Table 1-3: Horisontal lines between the 4 welfare quality principles would ease the overview for the reader.
Yes, the table is however in the design of the format required by Animals. Maybe Animals can adjust the table in the editing part.
Section 2.3 line 283: Suggest to mentioned the total number of ABM’s (also mentioned in abstract) for each species here – or maybe give them numbers in Table 1-3 – this makes it easier for the reader to keep track of differences.
The total number of ABMs per animal category are mentioned at the start of paragraph 3.1 and also in 3.3. We are not sure to which number of ABMs the reviewer refers in this comment. L283 refers to paragraph 3.2. Animal-based measures at the slaughterhouse recorded by the RA. We can not give an exact number of the ABMs used by the RA, because the work instructions only give examples and the descriptions are relatively vague.
Line 406-431: In the description of the principles of technologies only a few details is given, e.g. camera, but it would be nice if the technology behind could be elaborated a bit more.
This comment is not clear to the authors. L406-431 refer to paragraph 4.1. Existing technologies and AM and PM inspection. However, in this paragraph not many technologies are mentioned. Can the reviewer elaborate on this comment? To what kind of detail should the technology behind be described? For instance the mention of using still or moving images and a camera? Does this comment refer to paragraph 3.3 The use of sensor technology and AI to record animal-based measures?
Line 469 and 471: Not sure what is meant by the two parentheses: (..)
Text from the legislation is omitted. Brackets should be used instead of parentheses. This is adjusted. See L487 and 489
Discussion
Nice and thorough discussion.
The previous mentioned two levels of comparison could be elaborated more in discussion, i.e.: 1) sensor data compared to observations at slaughter house and 2) Relation between sensor data/observations and the actual animal welfare.
Thank you for this nice suggestion. This is indeed a relevant topic. An extra paragraph (5.2.) is added in the discussion on the comparison of sensor data and human observations at the slaughterhouse.
The relation between sensor data/observations and the actual animal welfare is added in L697- 700. A system of sensor technology gives no information on the actual welfare status of the animal as this is a combination of many factors.

Reviewer 2 Report
This manuscript regards the important topic of assessing animal welfare at the slaughterhouse through the evaluation of the so-called Animal-based Measure by new technologies based on Sensor Technology and Artificial Intelligence. Overall, the manuscript is fairly written and the author's efforts to address the topic are evident. However, I believe that there is some room for improvement especially in the Introduction section, which is a bit chaotic and hard to read.
The authors may find some points that need to be addressed prior to publication:
Simple summary:
The first sentence is hard to understand, please rephrase it.
Abstract
The formatting of the abstract is incorrect, as no separate paragraphs should be present.
Introduction
L45: Add reference(s) after ‘times a year’
L52: Rephrase (two times the word inspection was used)
L61: The PM inspection is exclusively visual only for pigs, poultry, and certain categories of cattle, please amend.
L64: The slaughterhouse is a sentinel point, the meat inspection per se is an activity carried out in a slaughterhouse, please amend.
L64-L77: I can’t fully understand the reasons why the authors have addressed in this way the issue concerning transport and on-farm improper welfare. I would just address the reason why the slaughterhouse is a convenient point to evaluate retrospective indicators of animal welfare.
L82-L83: The authors should add also that ABMs are objective, reliable, and measurable measures that can be assessed on an animal, hence why are preferred with respect to the not-ABMs.
L141-L168: I would include this section as a separate chapter with a separate Title, as this looks to me as a proper description of the application of Artificial Intelligence. The authors should include this new chapter within the description of the topic as per L170-L176.
Methods:
2.2. Please change to ‘Analysis of the Literature’
L245: Please specify the skills that make these scientists experts in the field (address their skills).
L262-L266: Redundant, please delete this part.
Table 2: Injuries (pain): The authors have not included any references when it comes to the assessment of bruises in cattle at the slaughterhouse, although some research has been done in this field. Examples are here reported (DOI):
10.3390/ani12151997
10.1002/vms3.2
10.5713/ajas.19.0804
L313: The Regulation is 2019/627, please amend.
L314: ‘broilers are inspected on carcass condemnations’, this sentence is not clear, please amend.
Minor editing of English language required
Author Response
Dear reviewer,
Thank you very much for taking the time to review this manuscript. Please find the detailed responses below and the corresponding revisions/corrections in track changes in the attachement.
The comments of the reviewer are in bold. Our response below. The line numbers refer to the line numbers in the document with track changes in the attachment.
Kind regards,
Simple summary:
The first sentence is hard to understand, please rephrase it.
Sentence is adjusted. See L9
Abstract
The formatting of the abstract is incorrect, as no separate paragraphs should be present.
Adjusted.
Introduction
L45: Add reference(s) after ‘times a year’
Several references are added. See L46
L52: Rephrase (two times the word inspection was used)
Second inspections is removed.
L61: The PM inspection is exclusively visual only for pigs, poultry, and certain categories of cattle, please amend.
Text is adjusted, as indeed also palpation and incisions can be performed. See L63-66
L64: The slaughterhouse is a sentinel point, the meat inspection per se is an activity carried out in a slaughterhouse, please amend.
Text is adjusted. See L68
L64-L77: I can’t fully understand the reasons why the authors have addressed in this way the issue concerning transport and on-farm improper welfare. I would just address the reason why the slaughterhouse is a convenient point to evaluate retrospective indicators of animal welfare.
The text is adjusted and sentence removed. See L70-73
L82-L83: The authors should add also that ABMs are objective, reliable, and measurable measures that can be assessed on an animal, hence why are preferred with respect to the not-ABMs.
Sentence is adjusted. See L88.
L141-L168: I would include this section as a separate chapter with a separate Title, as this looks to me as a proper description of the application of Artificial Intelligence. The authors should include this new chapter within the description of the topic as per L170-L176.
As suggested by reviewer 3, headings are added to the introduction. There is now a separate heading ‘Sensor technology and artificial intelligence’ for this part. We hope this solution is also satisfactory for reviewer 2.
Methods:
2.2. Please change to ‘Analysis of the Literature’
Adjusted
L245: Please specify the skills that make these scientists experts in the field (address their skills).
Adjusted: scientific researchers, with peer-reviewed publications is added. See L260
L262-L266: Redundant, please delete this part.
Lines are removed and line 281-283 is adjusted.
Table 2: Injuries (pain): The authors have not included any references when it comes to the assessment of bruises in cattle at the slaughterhouse, although some research has been done in this field. Examples are here reported (DOI):
10.3390/ani12151997
10.1002/vms3.2
10.5713/ajas.19.0804
The table gives an overview of the ABMs that can be assessed at the slaughterhouse. The place in the table gives information about the phase in the animal’s life the ABM provides information. Bruises can indeed be assessed at the slaughterhouse. But the references are placed under transport, because the ABM provides information about the transport phase.
L313: The Regulation is 2019/627, please amend.
Thank you for this accurate and thorough feedback on this small, but important detail. It should indeed be 2019. Adjusted. See L330
L314: ‘broilers are inspected on carcass condemnations’, this sentence is not clear, please amend.
Text is adjusted. See L330-331 the carcasses of broilers are inspected on abnormalities

Reviewer 3 Report
The paper reviewed sensor technology in relation to the meat inspection in slaughterhouses. It is suggested to use use ABM's of animals at the farm, during transport and at the slaughterhouse. It presents an overview of the present systems and the oppertunities.
Abstract:
L29 Also using sensor technology an OV can be responsible.
Introduction:
Headings are missing.
L79-110: Missing is the EU legislation regarding the BO and AWO. Which is their role in the whole proces in relation to the OV?
M&M
Which years were searched using PubMed?
Discussion
Missing the relation between the OV and the management of the slaughyterhouse in solving the problem. More over other members in the processing chain can be involved.
L508-509: Under supervision of the OV's validated systems at the farm and lorry can be used when the legislation changed.
Conclusions
L712-713: Why is the focus on slaughterhouse (AM, PM)?
Author Response
Dear reviewer,
Thank you very much for taking the time to review this manuscript. Please find the detailed responses below and the corresponding revisions/corrections in track changes in the attachement.
The comments of the reviewer are in bold. Our response below. The line numbers refer to the line numbers in the document with track changes in the attachment.
Kind regards,
Abstract:
L29 Also using sensor technology an OV can be responsible
Not sure how to adjust this in the abstract and what the reviewer exactly indicates with this comment? The mentioned exception refers to the role of the OV and sensor technology, which Is emphasized further in the full text.
Introduction:
Headings are missing.
Headings are added in the introduction.
L79-110: Missing is the EU legislation regarding the BO and . Which is their role in the whole proces in relation to the OV?
The AWO and the slaughterhouse staff are primarily responsible for the animal welfare at the slaughterhouse. A sentences is added. The RA and OV only supervise. Since the focus of this review is on the RA we do not go in further detail of the AWO, slaughterhouse staff etc,
M&M
Which years were searched using PubMed?
No limitation on publication year was set. This is added.
Discussion
Missing the relation between the OV and the management of the slaughterhouse in solving the problem. More over other members in the processing chain can be involved.
The commercial available systems are used by the slaughterhouse to improve animal welfare at the slaughterhouse. Some examples are given in L386-391.
The opportunities for the RA to use the commercial systems are discussed in paragraph 5.5. The prerequisites (collaboration with the slaughterhouse) are mentioned in this paragraph as well. We hope that in this way the relation between the OV and the slaughterhouse is mentioned to the satisfaction of the reviewer.
L508-509: Under supervision of the OV's validated systems at the farm and lorry can be used when the legislation changed.
Yes, there a possibilities for on farm systems and the use by the RA as well. This is a very interesting topic for further research. Sensor technology on farm can also provide valuable information for the RA. But this topic deserves its own research and most options are left out in this review paper. Many on farm applications are wearables or require animal handling.
Conclusions
L712-713: Why is the focus on slaughterhouse (AM, PM)?
The scope of this review paper is the slaughterhouse, therefore the focus in the conclusion is on the use of sensor technology at the slaughterhouse. Options for on farm use by the RA are left out in this research. See also our response to the previous comment.
We decide to focus on the slaughterhouse for this review paper because the slaughterhouse is a sentinel point in the animal production chain from which animal welfare can be efficiently measured during slaughter as well as retrospectively during earlier phases in the production chain (during transport and on farm). Some indicators of animal welfare on farm are easier to measure at the slaughterhouse than on the farm itself. These indicators, for example, only will become visible during a PM inspection, such as stomach lesion. Furthermore, the welfare of animals from multiple farms can be assessed at a single location, since animals are brought to the slaughterhouse in large numbers from multiple farms and assembly centres. Also the RA is always present at the slaughterhouse.
